# Supervision Exists Everywhere: A Data Efficient Contrastive Language-Image Pre-training Paradigm

**Yangguang Li** [*1,], **Feng Liang**[* 2], **Lichen Zhao**[* 1], **Yufeng Cui**[1],
**Wanli Ouyang**[3], **Jing Shao**[1], **Fengwei Yu**[1], **Junjie Yan**[1]
[1]SenseTime Research
[2]The University of Texas at Austin
[3]University of Sydney
{liyangguang,zhaolichen,cuiyufeng}@sensetime.com
jeffliang@utexas.edu

## Abstract

Recently, large-scale Contrastive Language-Image Pre-training (CLIP) (Radford et al., 2021) has attracted unprecedented attention for its impressive zero-shot recognition ability and excellent transferability to downstream tasks. However, CLIP is quite data-hungry and requires 400M image-text pairs for pre-training, thereby restricting its adoption. This work proposes a novel training paradigm, Data efficient CLIP (DeCLIP), to alleviate this limitation. We demonstrate that by carefully utilizing the widespread supervision among the image-text pairs, our De-CLIP can learn generic visual features more efficiently. Instead of using the single image-text contrastive supervision, we fully exploit data potential through the use of (1) self-supervision within each modality; (2) multi-view supervision across modalities; (3) nearest-neighbor supervision from other similar pairs. Benefiting from these intrinsic supervision, our DeCLIP-ResNet50 can achieve 60.4% zero-shot top1 accuracy on ImageNet, which is 0.8% above the CLIP-ResNet50 while using $7.1\times$ fewer data. Our DeCLIP-ResNet50 outperforms its counterpart in 8 out of 11 visual datasets when transferred to downstream tasks. Moreover, Scaling up the model and computing also works well in our framework. Our code, dataset and models are released at: https://github.com/Sense-GVT/DeCLIP

## 1 Introduction

Over the last few years, pre-trained models have greatly revolutionized computer vision (CV) and natural language processing (NLP). The first wave of exploring pre-trained models took place in the field of CV. Deep convolutional neural nets (Krizhevsky et al., 2012; Simonyan & Zisserman, 2014; He et al., 2016) are pre-trained on well-labeled ImageNet (Deng et al., 2009) and then transferred to downstream CV tasks (Girshick et al., 2014; Long et al., 2015; Vinyals et al., 2015). Standardly, CV models are pre-trained to predict a fixed set of pre-defined object categories, e.g., 1000 classes in ImageNet. However, this supervised pre-training is hard to scale since we need arduous human labeling to specify new visual concepts.

When pre-training meets NLP, the intrinsic supervision within the natural language makes the pre-training more scalable (Devlin et al., 2018; Radford et al., 2019; Brown et al., 2020). Witnessing the progress in NLP, researchers use natural language supervision to learn visual features. The language-image pre-training can scale up to a very large size, benefiting from abundant image-text pairs on the Internet. For instance, CLIP (Radford et al., 2021) and ALIGN (Jia et al., 2021) adopt the contrastive loss to push the embedding of matched image-text pairs together while pushing those of non-matched pairs apart. They achieve prestigious performance by learning from an enormous dataset that contains 400M/1B image-text pairs. However, these methods also require huge storage and computing resources, which is not affordable for most laboratories and companies. We argue

---

[*]The first three authors contribute equally. The order is determined by dice rolling.

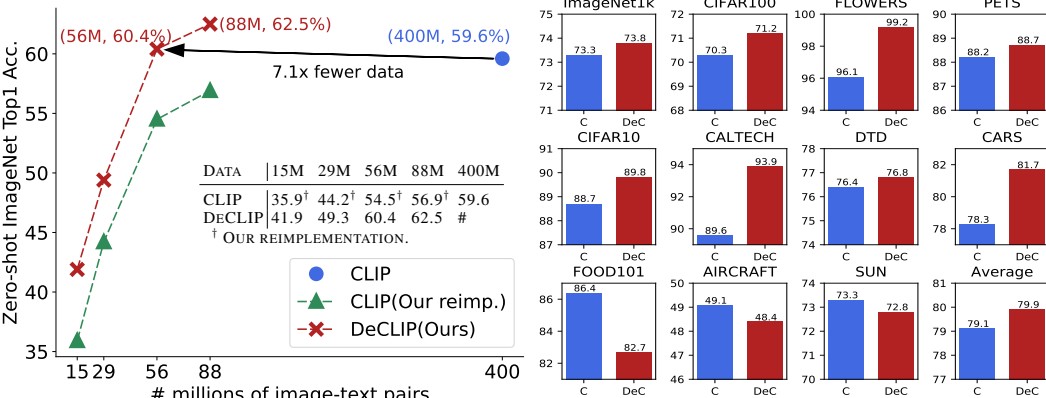

Figure 1: Zero-shot performance of CLIP-ResNet50 and our DeCLIP-ResNet50 when using different amounts of data. (88M, 62.5%) denotes the use of 88M data with top-1 accuracy 62.5% on the ImageNet-1K validation dataset. Our model has much better data efficiency.

Figure 2: Transfer the DeCLIP-ResNet50 (abbr. as DeC) and CLIP-ResNet50 (abbr. as C) to 11 downstream visual datasets using linear probe verification. Our DeCLIP achieves better results in 8 out of 11 datasets.

that these prior arts only use the single image-text contrastive supervision while overlooking the widespread supervision within the pairs, thus is inefficient.

Firstly, there underlies rich structural information within each modality itself (LeCun & Misra, 2021). We can tweak some words/pixels in a sentence/image while retaining a similar semantic meaning. This sort of self-supervision can be exploited to learn a more common-sense representation for each modality (Devlin et al., 2018; He et al., 2020; Chen et al., 2020a). Moreover, inspired by contrasting multi-crops in an image (Caron et al., 2020), we further extend the multi-view [1] supervision into our multi-modality setting. Specifically, each image is paired with multiple textual descriptions obtained via stochastic augmentations, vice versa. The benefit is intuitive: this auxiliary multi-view supervision brings more invariant and robust information.

Besides these overlooked supervision, we propose a novel nearest-neighbor (NN) supervision from other similar pairs. This NN supervision is mainly based on the intuition that one image is likely to have other similar text descriptions among the dataset. As shown in right figure, the image with the text `'going to see a lot of vintage tractors this week'` can also be described by `'vintage at tractors a gathering'`. For this reason, we sample the NN in the embedding space and utilize them as additional supervisory signals. Aggregating these supervision leads to our novel training paradigm DeCLIP, which stands for **D**ata **e**fficient **C**ontrastive **L**anguage-**I**mage **P**retraining.

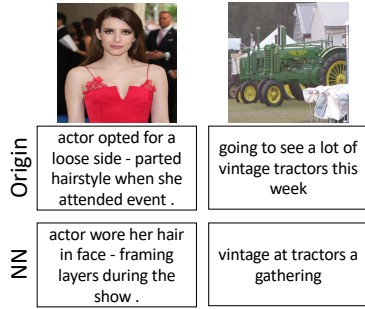

Figure 3: Examples of Nearest Neighbor from Conceptual Captions dataset.

Extensive experiments show the effectiveness and efficiency of our DeCLIP. As shown in Fig. 1, with a ResNet50 image encoder and a Transformer text encoder, our model can achieve 60.4% zero-shot top1 accuracy on ImageNet, which is 0.8% above the CLIP-ResNet50 while using 7.1× fewer data. Using only 88M image-text pairs, our best ResNet50/ViT-B32 models boost the zero-shot performance to 62.5% and 66.2%, nearly 3.0% higher than the best number reported for these two architectures. We further verify the transferability of our models on downstream tasks. As indicated in Fig. 2, our DeCLIP-ResNet50 outperforms its counterpart in 8 out of 11 visual datasets. Moreover, Scaling up the model and computing also works well in our framework. Using 4.5× fewer data, our DeCLIP-RegNetY-64GF achieves 73.7% zero-shot ImageNet top1 accuracy, which is on-pair with CLIP-R50×64. Pre-trained models, code, and datasets shall be released to the community. The contributions are summarized as follows:

---

[1]View is originally a visual concept. For simplicity, we use the same term for language.

- To the best of our knowledge, this is the first work to study self-supervision and cross-modal multi-view supervision in the million-scale image-text pre-training task. Our work opens a new direction to fully exploit the intrinsic supervision within the multi-modal data instead of scaling up data naively.

- We propose novel cross-modal Nearest-Neighbor Supervision (NNS) to harness information from other similar pairs. The NNS can also be regarded as a semantic-level augmentation.

## 2 RELATED WORK

### 2.1 PRE-TRAINED MODELS

The critical idea of pre-training is to first extract general knowledge implicitly from the massive amount of data and then transfer the knowledge to versatile downstream tasks (Han et al., 2021). Big NLP models (Devlin et al., 2018; Brown et al., 2020) yield unprecedented performance via learning from tremendous language data over the Internet and labor-free supervision within the language itself. In the field of CV, supervised pre-training on ImageNet is still the standard practice. While achieving great success on downstream CV tasks (Girshick et al., 2014; Long et al., 2015; Vinyals et al., 2015) , this supervised manner is hard to scale. To address this challenge, our DeCLIP learns directly from image-text pairs that are abundant across the Internet. More importantly, by exploiting the widespread supervision within the pairs, our DeCLIP is more data-efficient than the prior art.

### 2.2 SUPERVISION WITHIN DATA

**Language supervision** Joulin et al. (2016); Gomez et al. (2017); Zhang et al. (2020); Sariyildiz et al. (2020); Desai & Johnson (2021) demonstrate the effectiveness of learning transferable visual features from language supervision. Pioneering work CLIP (Radford et al., 2021) and ALIGN (Jia et al., 2021) achieve prestigious performance via learning from 400M/1B image-text pairs. We are following these two works to improve their data efficiency.

Relevant concurrent work including: SLIP Mu et al. (2021) introduces self-supervision to CLIP. FILIP Yao et al. (2021) leverages the finer-grained alignment between image patches and textual words. OTTER Wu et al. (2021); Cheng et al. (2021) uses online entropic optimal transport to find a soft image-text match as labels to mitigate the noise within the dataset.

**Visual self-supervision** Our work is also highly related to self-supervised learning (SSL) (LeCun & Misra, 2021). Contrastive learning, as a pretext task of SSL, has achieved remarkable success in visual representation learning (He et al., 2020; Chen et al., 2020a; Caron et al., 2020; Grill et al., 2020; Chen et al., 2020a). Researchers also extend contrastive learning into multi-modal settings (Yuan et al., 2021). However, it is only limited to a small COCO dataset (Yuan et al., 2021).

**Nearest-neighbor supervision** Recently, researchers have exploited nearest-neighbor supervision to learn visual features (Dwibedi et al., 2021; Van Gansbeke et al., 2021). They find that using nearest-neighbor as positive samples in the contrastive loss improves the performances on multiple downstream tasks. However, they mainly focus on the single visual modality pretraining on relatively small datasets, such as ImageNet. We propose novel nearest-neighbor supervision for multi-modal learning to harness information from other similar pairs.

### 2.3 MULTI-MODAL LEARNING

Most vision-language models Chen et al. (2020b); Lu et al. (2019); Li et al. (2020) use a bunch of cross-modal transformers to fuse and align the information between text and image. These methods either need an off-the-shelf object detector to extract region features or dedicated cross-modal transformer layers, significantly hindering their scalability. Our DeCLIP, by contrast, uses a simple yet effective two-tower framework with multi-modal interaction only at the top. Moreover, this series of models (Radford et al., 2021; Jia et al., 2021; Huo et al., 2021) can perform zero-shot recognition, adapting to new categories with no seen labeled data. Shen et al. (2021) also shows that the pre-trained CLIP model can significantly benefit the downstream VQA and image caption tasks. Our DeCLIP is supposed to be compatible with more modalities, e.g., acoustic signals (Akbari et al., 2021). More modalities included, more correlated supervision are expected to be exploited.

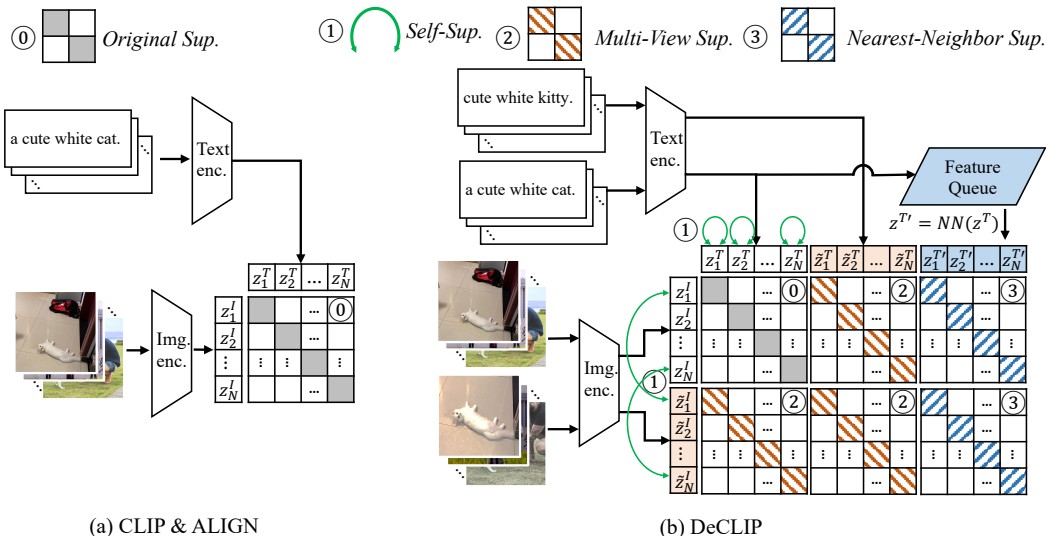

Figure 4: (a) CLIP and ALIGN jointly train an image encoder and a text encoder to predict the correct pairings of a batch of (image, text) training examples. (b) Our DeCLIP overview. ① means Self-Supervision(SS). For image SS, we maximize the similarity between two augmented views of the same instance. For text SS, we leverage Masked Language Modeling(MLM) within a text sentence. ② represents cross-modal Multi-View Supervision(MVS). We first have two augmented views of both image and text, then contrast the $2 \times 2$ image-text pairs. ③ indicates Nearest-Neighbor Supervision(NNS). We sample text NN in the embedding space to serve as additional supervision. The combination of the three supervision leads to efficient multi-modal learning.

## 3 APPROACH

In this section, we first revisit CLIP and denote some basic concepts, such as image-text contrastive supervision (i.e., the InfoNCE loss). Next, we present the overview of our DeCLIP framework. Then we introduce every auxiliary supervision: Self-Supervision(SS), Multi-View Supervision(MVS), and Nearest-Neighbor Supervision(NNS).

### 3.1 REVISITING CLIP

Contrastive Language-Image Pre-training (CLIP) (Radford et al., 2021) aims to learn directly from the raw text about images. They use a dual-encoder architecture as in Fig. 4(a). The model consists of an image encoder (e.g., CNN (He et al., 2016) or ViT (Dosovitskiy et al., 2020)) and a text encoder(e.g., Transformer (Vaswani et al., 2017) or its variants (Radford et al., 2019)), with a multimodal interaction at the top. The image and text features are projected to the same dimension and followed by L2 normalization before interaction. At the training phase, a contrastive objective pushes the embeddings of matched image-text pairs together while pushing those of non-matched pairs apart. In a batch of $N$ image-text pairs $\{(\boldsymbol{x}_i^I, \boldsymbol{x}_i^T)\}$, we denote $\boldsymbol{x}_i^I$ and $\boldsymbol{x}_i^T$ as image and text of the $i_{th}$ pair. Let $\boldsymbol{z}_i^I$ and $\boldsymbol{z}_j^T$ be the normalized embedding of the $i_{th}$ image and $j_{th}$ text, respectively. CLIP uses InfoNCE loss (Oord et al., 2018). The loss for the image encoder can be denoted as Eq. 1.

$$L_I = -\frac{1}{N} \sum_{i=1}^{N} \log \frac{\exp(\text{sim}(\boldsymbol{z}_i^I, \boldsymbol{z}_i^T)/\tau)}{\sum_{j=1}^{N} \exp(\text{sim}(\boldsymbol{z}_i^I, \boldsymbol{z}_j^T)/\tau)} \tag{1}$$

Here, the similarity function $\text{sim}(,)$ is measured by dot product, and $\tau$ is a learnable temperature variable to scale the logits. We have a symmetrical loss for image and text encoder, thus the overall loss function $L_{CLIP}$ is the average of $L_I$ and $L_T$.

At the test phase, the learned text encoder synthesizes a zero-shot linear classifier by embedding the arbitrary categories of the test dataset. Because it is rare in the dataset that image caption is just

a single word, CLIP uses prompts to make up the context of the category {label}, such as "a photo of a {label}". Unless otherwise specified, we use the same prompt engineering and ensembling techniques as CLIP. Details can be found in Appendix E.

## 3.2 OVERVIEW OF DeCLIP

As shown in Fig. 4(b), our DeCLIP has three additional supervisory signals.

①We first use existing methods to exploit image and text Self-Supervision (SS) within its modality. For image SS, we adopt the simple yet effective SimSiam (Chen & He, 2021). The objective is to maximize the similarity between two augmented image features. For text SS, we adopt the most widely used Masked Language Modeling (MLM) (Devlin et al., 2018) as the pre-text task. We believe other kinds of self-supervised learning algorithms (e.g. MoCo (He et al., 2020), SimCSE (Gao et al., 2021) are orthogonal with our framework.

②While SS only focuses on a single modality, we further propose cross-modal Multi-View Supervision (MVS). We apply stochastic data augmentations for both images and texts, resulting in two correlated views[1] of each example. Then, the image-text contrastive loss is calculated for all the $2 \times 2$ pairs. Worth mentioning, the original CLIP does not use text augmentations and only uses random square crop image augmentations, thereby is data-hungry. The extension is instinctive and straightforward. Specifically, we contrast the $2 \times 2$ pairs and resulting in $3\times$ more additional supervision.

③We also propose novel Nearest-Neighbor Supervision (NNS) mined in embedding space to make better use of similar text descriptions among the dataset. In detail, we maintain a first-in-first-out feature queue that is representative of the whole data distribution. We use the nearest-neighbor search in embedding space to get the semantically similar text descriptions. Then we use the image-text contrastive loss to get additional supervision.

## 3.3 SUPERVISION EXISTS EVERYWHERE

**Self-Supervision within each modality** Following Sim-Siam (Chen & He, 2021) (depicted in Fig 5(a)), we first have two augmented views $(x^I, \tilde{x}^I)$ for each image. These two views are sent to the image encoder (weights are shared between views). We also use the popularized nonlinear predictor module, which is typically a 2-layer MLP, to improve the representation quality in the encoder (Chen et al., 2020a). The objective is to maximize the similarity between $\tilde{z}^I$ and $p^I$, which is a negative cosine similarity in this paper. To avoid the trivial "collapsing" solution, we follow (Chen & He, 2021) to adopt a stop-grad technique.

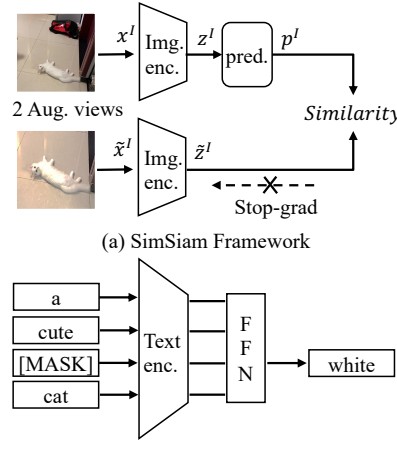

(a) SimSiam Framework

(b) Masked Language Modeling

Figure 5: Self-Supervision with each modality. We adopt SimSiam and MLM for image and text SS.

As shown in Fig. 5(b), we follow the method in BERT (Devlin et al., 2018) for our text self-supervision. In detail, we first randomly choose $15\%$ of all tokens in each sequence. Then the token is replaced with (1) the [mask] token $80\%$ of the time (2) a random token $10\%$ of the time (3) the unchanged token $10\%$ of the time. Then, the output of the language module for the corresponding token is used to predict the original token with cross-entropy loss.

**Multi-View Supervision** The authors only contrast the original text (w/o augmentation) with a single 'global view' of the image in the original CLIP. However, the text annotation of the image might not describe the whole picture, instead depicts a small local view of this image. For instance, as shown in the image with the text "a cute white cat" in Fig. 4, the central concept (cat) only occupies a small part of the picture. To mitigate this discrepancy, we get a closer look at the local region and utilize it as our auxiliary supervision, as shown in the augmented view in Fig. 4(b). This intuitive idea is akin to the successful Multi-crop transformation (Caron et al., 2020; Van Gansbeke et al., 2021) in image SSL. We further extend it into the multi-modal

setting. More specifically, we reuse the two image views introduced in SS, which contains the `RandomResizedCrop` policy to obtain a small local view. For text, as our goal is to understand the overall semantic meaning of a sentence, we adopt text classification augmentation EDA (Wei & Zou, 2019) to generate two text views. Besides the original contrastive loss between $(z^I, z^T)$, we can contrast $(z^I, \tilde{z}^T)$, $(\tilde{z}^I, z^T)$ and $(\tilde{z}^I, \tilde{z}^T)$, leading to $3\times$ diverse and high-quality additional supervision. More conveniently, they are naturally compatible with the image-text contrastive loss as denoted in Eq. 1.

**Nearest-Neighbor Supervision** As shown in Fig 3, one image is likely to have other similar text descriptions among the dataset. To harness the information from other pairs and go beyond single pairs, we propose using nearest-neighbor (NN) to obtain more diverse supervision. More formally, we aim to find the NN feature $z^{T'}$ of text feature $z^T$ in the embedding space. The distance between two features could be measured by a simple cosine similarity. It is infeasible to search NN in the whole million-scale dataset. Thus, we maintain a FIFO queue $Q$ to simulate the whole data distribution. The size of $Q$ is 64K in our implementation. As shown in Fig. 6, we further get the contrastive loss between $(z^I, z^{T'})$. Since there are two augmented image features, we also calculate the contrastive loss between $(\tilde{z}^I, z^{T'})$. Fortunately, NNS is also compatible with Eq. 1.

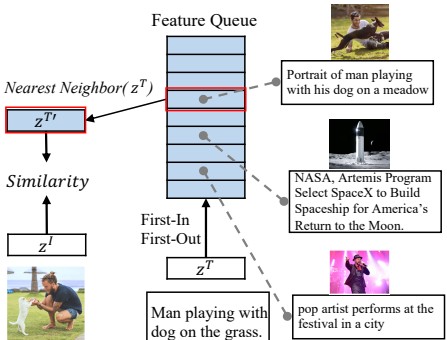

Figure 6: Nearest-Neighbor Supervision. $z^{T'}$ is the NN of feature $z^T$ in the embedding space. $z^{T'}$ will serve as an additional objective for $z^I$. We use the feature-level nearest neighbor for the text descriptions as the supervision.

In summary, we denote $L_{ISS}$ and $L_{TSS}$ as the loss function of image SS and text SS, respectively. $L_{MVS}$ is multi-view loss, and $L_{NNS}$ is nearest-neighbor loss. We have the overall loss function of our DeCLIP as in Eq. 2.

$$L_{DeCLIP} = (1 - \alpha - \beta - \gamma)L_{CLIP} + \alpha(L_{ISS} + L_{TSS}) + \beta L_{MVS} + \gamma L_{NNS} \qquad (2)$$

## 4 EXPERIMENTS

### 4.1 DATASETS

**Pre-training datasets** We summarize our pre-training dataset as in Tab. 1. Our DeCLIP full data consists of two parts: open-source data and web-crawled data. The open-source data comes from three different datasets: Conceptual Captions (CC3M) (Sharma et al., 2018), Conceptual 12M (CC12M) (Changpinyo et al., 2021), and YFCC (Thomee et al., 2016). Worth mentioning, due to the download failure or non-English caption, we do not obtain the complete data for these datasets. We further use the YFCC15M query to crawl about 59M filtered

Table 1: Details of DeCLIP pre-training datasets.

| DATASET | TRAINING SIZE |
|---|---|
| CLIP (RADFORD ET AL., 2021) | 400M |
| ALIGN (JIA ET AL., 2021) | 1.8B |
| CC (SHARMA ET AL., 2018) | 3M |
| CC-12M (CHANGPINYO ET AL., 2021) | 11M |
| YFCC (THOMEE ET AL., 2016) | 15M |
| **DeCLIP OPEN-SOURCE DATA** | 29M |
| DeCLIP WEB-CRAWLED DATA | 59M |
| **DeCLIP FULL DATA** | 88M |

web data on the Internet, together with open-source data to form the 88M DeCLIP full pre-training dataset. More details can be seen in Appendix C.

**Downstream datasets** We assess our model performances in a wider variety of distributions and tasks. Following the CLIP (Radford et al., 2021), we evaluate the image encoder transferability on 11 widely used downstream datasets, such as Food-101, CIFAR-10, etc. To conduct a fair comparison, the metric and division for each dataset that can be collected are consistent with those of CLIP. More details of downstream datasets can be found in Tabel 6 of Appendix D.

Table 2: Zero-shot top1 accuracy on ImageNet. Our DeCLIP shows great data-efficency.

| METHOD | IMAGE ENCODER | # PARAMS | TRAINING SIZE | ZERO-SHOT TOP1 ACC. |
|---|---|---|---|---|
| CLIP[†] | RESNET50 | 24M | 88M | 56.9 |
| CLIP | RESNET50 | 24M | 400M | 59.6 |
| DECLIP | RESNET50 | **24M** | **88M($\downarrow$ 4.5$\times$)** | **62.5($\uparrow$ +2.9)** |
| CLIP | RESNET101 | 42M | 400M | 62.2 |
| CLIP[†] | VIT-B/32 | 88M | 88M | 57.4 |
| CLIP | VIT-B/32 | 88M | 400M | 63.2 |
| DECLIP | VIT-B/32 | **88M** | **88M($\downarrow$ 4.5$\times$)** | **66.2($\uparrow$ +3.0)** |
| CLIP | RESNET50$\times$64 | 291M | 400M | 73.6 |
| DECLIP | REGNETY-64GF | **276M** | **88M($\downarrow$ 4.5$\times$)** | **73.7($\uparrow$ +0.1)** |

[†] OUR REIMPLEMENTATION.

## 4.2 EXPERIMENTS SETUP

**Network architectures** Following CLIP, we first consider two different architectures for the image encoder: a modified version of ResNet50 (Radford et al., 2021) and ViT-B/32 (Dosovitskiy et al., 2020). The text encoder is a Transformer (Vaswani et al., 2017) with the architecture modifications described in Radford et al. (2019). The image and text features are projected to the same 1024 dimension, followed by L2 normalization before interaction. Benefiting from the rapid progress of large CV models, we further scale up our model. Our largest model is a RegNetY-64GF (Radosavovic et al., 2020; Goyal et al., 2021) image encoder with a BERT (Devlin et al., 2018) text encoder, which is on-pair with the largest CLIP-R50$\times$64 model.

**Pre-training setup** For a fair comparison with CLIP, we train our DeCLIP-ResNet50 and DeCLIP-ViT-B/32 from scratch for 32 epochs. Unless otherwise specified, we use full data, i.e., 88M image-text pairs, to obtain the best performance. The input resolution of the image encoder is $224 \times 224$, and the maximum context length of the text encoder is 76. The learnable temperature parameter $\tau$ is initialized to 0.07. The loss weights of additional supervision $\alpha$, $\beta$ and $\gamma$ are all set to 0.2. More details can be found in Appendix C.

**Downstream evaluation setup** We evaluate our model transferability by performing linear classification on frozen features, i.e., the pre-trained image encoder is fixed and serves as a feature extractor. After feature extraction, we train the linear classifier with the L-BFGS optimizer as the same in Radford et al. (2021).

## 4.3 MAIN RESULTS

**Zero-shot recognition on ImageNet** After pre-training, we use natural language to refer to visual concepts enabling the zero-shot ability of our model. As shown in Fig. 1, our DeCLIP-ResNet50 consistently outperforms the CLIP-ResNet50 across all dataset sizes. When the data amount reaches 56M (29M open-source + 27M web-crawled), our model can achieve 60.4% accuracy, 0.8% above the CLIP-ResNet50, while using 7.1$\times$ fewer data. As described in Tab. 2, with our full data, our best ResNet50/ViT-B32 models boost the zero-shot performance to 62.5% and 66.2%, nearly 3.0% higher than the best number reported for these two architectures. Moreover, our DeCLIP-ResNet50 is even 0.3% better than CLIP-ResNet101, revealing the effectiveness and efficiency of our framework. Scaling up model capacity works in our framework as well. Our biggest DeCLIP-RegNetY-64GF achieves 73.7% accuracy, which is 0.1% above CLIP ResNet50$\times$64 with fewer parameters.

**Downstream evaluation results** We report our linear probe performance on 11 downstream datasets in Tab. 3. Our DeCLIP-ResNet50 outperforms its CLIP counterpart in 8 out of 11 datasets, with a 0.8% average improvement. There are some datasets that our models perform worse than CLIP models, such as SUN and Food101. We conjecture that this is caused by the different distribution of the pre-trained dataset. Interestingly, our ResNet50 and ViT-B/32 models might have distinct performance on several datasets, such as Pets and Aircraft. We infer that these two types

Table 3: Linear probe performance on 11 downstream datasets. There are some abbreviations. C10/100 is CIFAR10/100. F101 is Food101. Flow is Flowers. Cal is Caltech. Air is Aircraft. IN is ImageNet. Our DeCLIP models achieve higher average accuracy over 11 datasets.

| Model | PETS | C10 | C100 | SUN | F101 | FLOW. | CARS | CAL. | AIR. | DTD | IN | AVG. |
|---|---|---|---|---|---|---|---|---|---|---|---|---|
| CLIP-ResNet50[†] | 85.1 | 87.3 | 65.0 | 71.3 | 80.8 | 98.2 | 77.2 | 89.6 | 44.8 | 71.0 | 70.8 | 76.5 |
| CLIP-ResNet50 | 88.2 | 88.7 | 70.3 | **73.3** | **86.4** | 96.1 | 78.3 | 89.6 | **49.1** | 76.4 | 73.3 | 79.1 |
| DeCLIP-ResNet50 | **88.7** | **89.8** | **71.2** | 72.8 | 82.7 | **99.2** | **81.7** | **93.9** | 48.4 | **76.8** | **74.0** | **79.9(↑ +0.8)** |
| CLIP-ViT-B/32[†] | 85.3 | 91.6 | 74.2 | 72.3 | 80.8 | 97.9 | 77.6 | 93.2 | 47.0 | 72.5 | 70.3 | 78.4 |
| CLIP-ViT-B/32 | **90.0** | 95.1 | 80.5 | **76.6** | **88.8** | 96.9 | **81.8** | 93.0 | 52.0 | 76.5 | **76.1** | 82.5 |
| DeCLIP-ViT-B/32 | 89.2 | **96.5** | **84.7** | 75.0 | 85.0 | **99.1** | 81.6 | **94.8** | **53.5** | **78.5** | 75.3 | **83.0 (↑ +0.5)** |

[†] Our reimplementated CLIP model trained on 88M data.

Table 4: Ablation on additional supervision. SS/MVS/NNS denotes Self-Supervision, Multi-View Supervision and Nearest-Neighbor Supervision, respectively.

| CLIP | MVS | SS | NNS | ZERO-SHOT |
|---|---|---|---|---|
| ✓ | × | × | × | 20.6 |
| ✓ | ✓ | × | × | 24.8(↑ +4.2) |
| ✓ | ✓ | ✓ | × | 25.4(↑ +4.8) |
| ✓ | ✓ | ✓ | ✓ | **27.2(↑ +6.6)** |

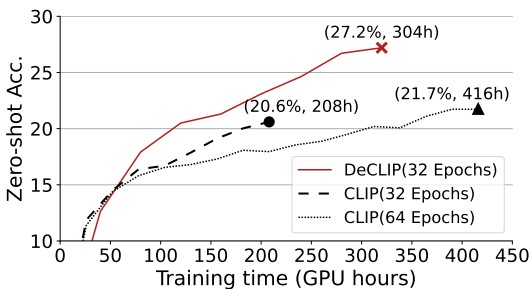

Figure 7: Ablation on pre-training cost on CC3M dataset. The proposed method performs better with less training time.

of neural networks might have different data preferences, i.e., different feature extraction capacities when they meet the same data.

## 4.4 ABLATION STUDY

**Ablation on additional supervision**   In order to understand the effectiveness of each additional supervision, we conduct an ablation study as indicated in Tab. 4. We follow the DeCLIP-ResNet50 protocol in the pre-training setup except for a smaller 1024 batch size on a smaller CC3M dataset. The single image-text contrastive supervision (CLIP) results in 20.6% zero-shot top1 accuracy on ImageNet. We can observe that MVS boost amazing 4.2% improvement over the original CLIP. As discussed in Sec. 3.3, the benefits might come from two sides: 1). the MVS can look at a small local view of an image which might be a better fit with the text description.  2).  $2 \times 2$ augmented views can provide $3\times$ diverse and high-quality additional supervision, thus leading to more robust representations. SS can further contribute an additional 0.6% improvement on the basis of MVS. We believe SS could bring more improvements with proper dedicated SSL methods. NNS further brings 1.8% improvement on the high basis of SS. We will discuss more about NNS at Sec. 4.5. On the YFCC15M dataset, our DeCLIP using full additional supervision gets significant effect improvement, Appendix F shows the details.

**Ablation on training cost**   Since we need to encode twice for each image-text pair, we admit our DeCLIP needs a higher training cost than CLIP. Regarding the training time, one DeCLIP iteration equals $1.5\times$ CLIP iteration. In this ablation, we train the original CLIP-ResNet50 longer than our DeCLIP-ResNet50. As shown in Fig. 7, longer 64 epochs training can bring about 1.1% improvement. However, our model has 27.2% top1 accuracy, which is still 5.3% higher than the time-equivalent CLIP. It reveals that our framework can extract richer and more representative features through our proposed supervision. We also include memory usage ablation in Appendix F.

## 4.5 ANALYSIS

**Class activation maps**   We try to understand what renders our DeCLIP effective. As shown in Fig. 8, we visualize the class activation maps (CAM) (Zhou et al., 2016) of different models trained

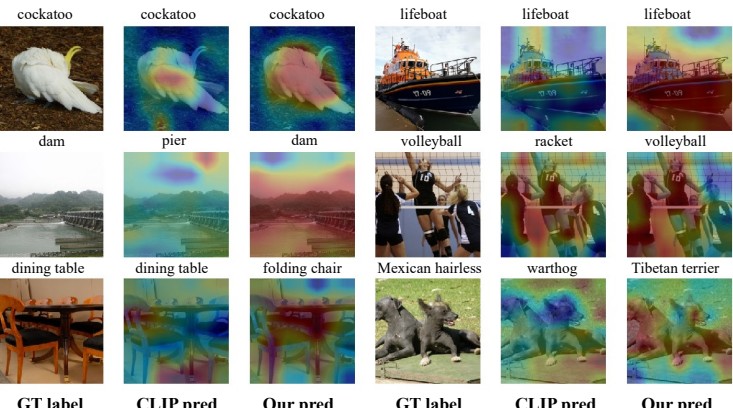

Figure 8: Class activation maps (CAM) for the CLIP vs. our DeCLIP model trained on the YFCC dataset. The CAMs of our model segment the complete object, while the CLIP model only looks at a few components.

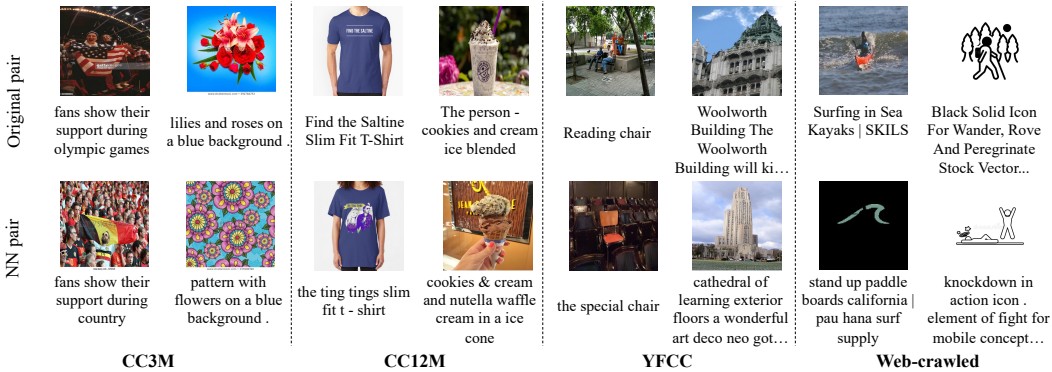

Figure 9: Nearest neighbor samples from different datasets. As we can see, the NN pair is very similar to the original pair, thus it can provide high-quality supervision.

on the YFCC dataset. The results validate that the proposed method learns more representative features with the aid of multiple supervision.

**Nearest neighbor samples** In Fig 9, we show some nearest neighbor (NN) samples from different datasets. The first row is the original image-text pair, while the second row is the NN pair. In general, we can see that the texts have similar intellectual meanings. Therefore, the NN pair can provide high-quality supervision. When taking datasets into consideration, the matching performs very well in well-filtered datasets, such as CC3M and CC12M. This matching might be a little worse in more noisy datasets, such as YFCC and web-crawled, but it can still provide some beneficial guidance.

## 5 CONCLUSION

This paper introduces DeCLIP, a Data efficient Contrastive Language-Image Pre-training paradigm. Our goal is to learn visual representations through the use of broader and scalable supervision. Specifically, instead of using the single image-text contrastive supervision, we fully exploit data potential through the use of (1) self-supervision within each modality; (2) multi-view supervision across modalities; (3) nearest-neighbor supervision from other similar pairs. Experimentally, De-CLIP shows superior effectiveness and efficiency with different types of neural nets(CNN and ViT) and different amounts of data. We hope our work could bring insights about exploiting the multi-modal data.

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

## A  PSEUDO CODE OF DeCLIP

The training pseudo code of DeCLIP is as follows:

---
**Algorithm 1** DeCLIP

---
**Input:** $I, \tilde{I}, T, \tilde{T}, image\_encoder, text\_encoder, Feature\_Queue$
1: **function** BATCH-UPDATING($I, \tilde{I}, T, \tilde{T}$)
2:     # Get The Features of The Current Batch.
3:     $I_f, \tilde{I}_f \leftarrow image\_encoder(I), image\_encoder(\tilde{I})$
4:     $T_f, \tilde{T}_f \leftarrow text\_encoder(T), text\_encoder(\tilde{T})$
5:     $T_{NN} \leftarrow$ NEAREST-NEIGHBOR($Feature\_Queue, T_f$)
6:
7:     # Calculate The losses.
8:     $L_{CLIP} \leftarrow$ INFONCE-LOSS($I_f, T_f$)
9:     $L_{SS} \leftarrow$ IMAGE-SS-LOSS($I_f, \tilde{I}_f$) + TEXT-SS-LOSS($T_f$)
10:    $L_{MVS} \leftarrow$ INFONCE-LOSS($I_f, \tilde{T}_f$) + INFONCE-LOSS($\tilde{I}_f, T_f$) + INFONCE-LOSS($\tilde{I}_f, \tilde{T}_f$)
11:    $L_{NNS} \leftarrow$ INFONCE-LOSS($\tilde{I}_f, \tilde{T}_{NN}$)
12:    $L_{DeCLIP} \leftarrow (1 - \alpha - \beta - \gamma)L_{CLIP} + \alpha L_{SS} + \beta L_{MVS} + \gamma L_{NNS}$
13:
14:    # Update The Network.
15:    $image\_encoder \leftarrow$ BACKWARD-UPDATE($image\_encoder, L_{DeCLIP}$)
16:    $text\_encoder \leftarrow$ BACKWARD-UPDATE($text\_encoder, L_{DeCLIP}$)
17:    $Feature\_Queue \leftarrow$ FIFO-UPDATE($Feature\_Queue, T_f$)
18: **end function**
19:
20: **function** IMAGE-SS-LOSS($I_f, \tilde{I}_f$)
21:    $z, \tilde{z} \leftarrow image\_encoder.proj(I_f), image\_encoder.proj(\tilde{I}_f)$
22:    $p, \tilde{p} \leftarrow image\_encoder.pred(z), image\_encoder.pred(\tilde{z})$
23:    $z, \tilde{z} \leftarrow z.detach(), \tilde{z}.detach()$
24:    # Calculate the Negative Cosine Similarity loss.
25:    $L_{Image-SS} \leftarrow NCS(p, \tilde{z})/2 + NCS(\tilde{p}, z)/2$
26:    **return** $L_{Image-SS}$
27: **end function**
28:
29: **function** TEXT-SS-LOSS($T_f$)
30:    # Get The Masking GT when performing the text-encoder.
31:    $W_f, W_{gt} \leftarrow T_f.word\_feat, T_f.mask\_id$
32:    $W_{pred} \leftarrow text\_encoder.pred(W_f)$
33:    # Calculate the Cross-Entropy loss.
34:    $L_{Text-SS} \leftarrow CE(W_{gt}, W_{pred})$
35:    **return** $L_{Text-SS}$
36: **end function**

---

## B  DATA AUGMENTATION

**Details of SS**  The image SS uses SimSiam (Chen & He, 2021) method as the image self-supervision. The prediction module is a 2-layer MLP, in which the hidden dimensions are 512 and output dimensions are 1024, the projection module is a 3-layer MLP, in which the hidden and output dimensions are both 1024. The text SS uses the Masked Language Model (Devlin et al., 2018) as the pretext task. In detail, we first randomly choose 15% of all tokens in each sequence. Then the token is replaced with (1) the `[mask]` token 80% of the time (2) a random token 10% of the time (3) the unchanged token 10% of the time.

**Image augmentations**  The augmentation policy includes: `RandomResizedCrop` with scale in [0.2,1.0] (Wu et al., 2018), `ColorJitter` containing {brightness, contrast, saturation, hue}

strength of {0.4, 0.4, 0.4, 0.1} with an applying probability of 0.8, `RandomGrayscale` with an applying probability of 0.2. Blurring augmentation (Chen et al., 2020a) has a Gaussian kernel with std in [0.1, 2.0], and `RandomHorizontalFlip`.

**Text augmentations**   We use the EDA (Wei & Zou, 2019) method as our text augmentation strategy, which contains three types of text augmentation strategies: synonym replacement, random swap, and random deletion. Each text will randomly select one of these three types for text augmentation.

## C   PRE-TRAINING DATASETS & IMPLEMENTATION DETAILS

**Open-source data.**   Conceptual Captions (Sharma et al., 2018) is a 3.3 M image caption open-source data. Due to the failure of the download link, we only download about 3M data(CC3M). Conceptual 12M (Sharma et al., 2018) contains approximately 12M of image-text pairs(CC12M), which is larger than the CC3M and covers a more diverse set of visual concepts. Also, due to the failure of the download link, we only download about 11M data. YFCC (Thomee et al., 2016), the Yahoo Flickr Creative Commons 100M data, is a dataset for vision language tasks. We download about 86.5M data from the YFCC website and use four filtering rules to filter the DeCLIP YFCC15M dataset to benchmark against CLIP YFCC15M. The four rules are: filtering data with damaged images, filtering data without the caption, filtering data with a caption English word ratio less than 0.8, filtering data with a caption only including one part of speech.

**Web-crawled data.**   We use the user tags and machine tags of YFCC15M to form a tag list, and use WordNet (Miller, 1995) to find synonyms for each tag in the tag list to form a synonym tag list. The tag list and synonym tag list form a query list. Then we use the query list to crawl images from the Internet, after filtering data with smaller images, filtering data with damaged images, filtering data without the caption, and filtering data with Chinese in the caption, we collect 59M web crawled data. Tabel 5 shows the source link of pre-training datasets, and Figure 10 shows some cases random sampled from each dataset.

Table 5: The source link of DeCLIP pre-training datasets.

| DATASET | DATASET API |
| --- | --- |
| CONCEPTUALCAPTIONS | HTTPS://AI.GOOGLE.COM/RESEARCH/CONCEPTUALCAPTIONS |
| CONCEPTUAL12M | HTTPS://GITHUB.COM/GOOGLE-RESEARCH-DATASETS/CONCEPTUAL-12M |
| YFCC | HTTP://PROJECTS.DFKI.UNI-KL.DE/YFCC100M |
| GOOGLE | HTTPS://WWW.GOOGLE.COM.HK |

**Implementation details**   We train DeCLIP-ResNet50 (abbr. as R50) and DeCLIP-ViT-B/32 (abbr. as V-B32) from scratch for 32 epochs. For R50, we use the FP16-SGD optimizer with the batch size of 10,240 (128×80). Starting with an 0.01 learning rate (lr), we first linearly increasing the lr to 0.2 (a.k.a warm-up) in one epoch. Then we use cosine anneal lr decay to decrease the lr. The weight decay is set to 0.0001. For V-B32, we use a hybrid FP16-AdamW-SGD optimizer with bacth size 10,240(128×80). For the ViT image encoder, we use AdamW optimizer with the lr warming up from 1e-4 to 1e-3 in one epoch. The weight decay is set to 0.05. For the text encoder, we use the SGD optimizer with the lr warming up from 1e-3 to 0.02 in one epoch. The weight decay is set to 1e-4.

**Pre-training cost**   Our R50 and V-B32 took 8/10 days to train on 80 V100 GPUs, respectively. Our largest DeCLIP-RegNetY-64GF took 21 days on 160 V100 GPUs, while the largest CLIP-R50×64 from  (Radford et al., 2021) spent 18 days on 592 V100 GPUs.

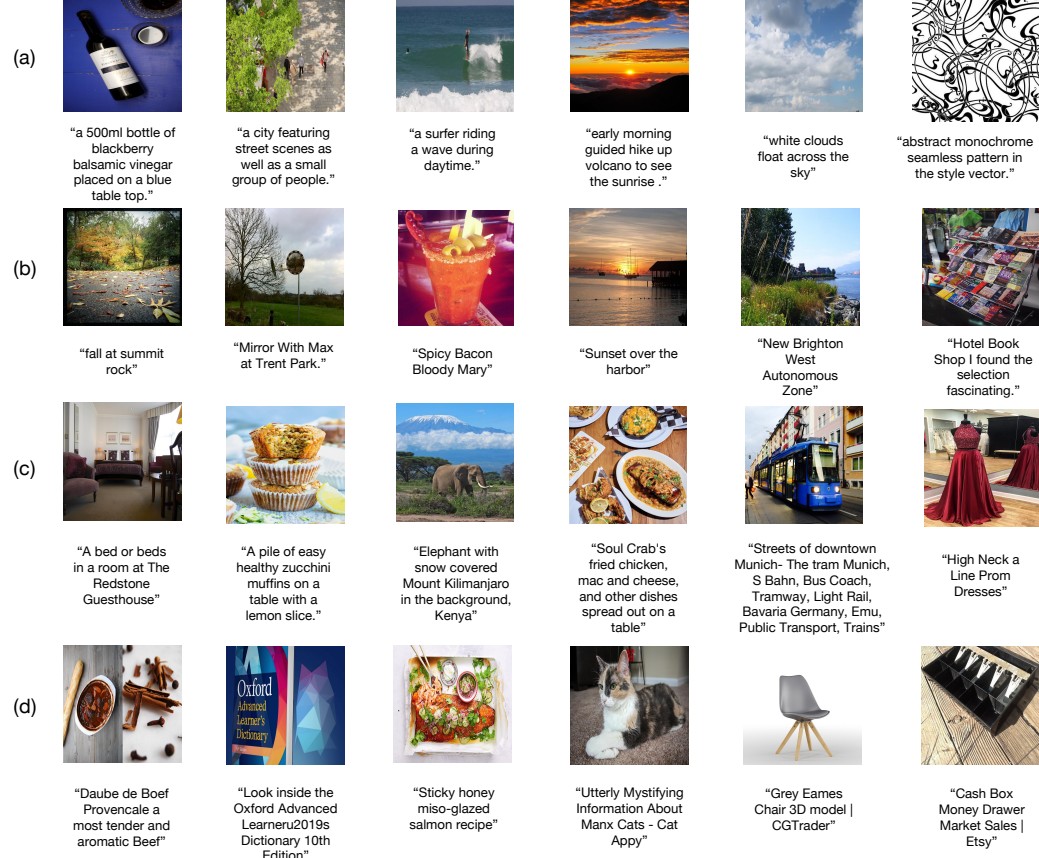

Figure 10: Example image-text pairs randomly sampled from the training dataset. (a) Conceptual Captions, (b) YFCC, (c) Conceptual 12M, (d) Web-crawled.

## D  DOWNSTREAM DATASETS & IMPLEMENTATION DETAILS

**Downstream data.**   We begin with the 12 datasets from the well-studied evaluation suite introduced by  Kornblith et al. (2019). Within these 12 datasets, Birdsnap can not be downloaded, and PASCAL VOC 2007 classification is replaced by more challenging ImageNet-1K, resulting in our 11 datasets. They are: Food-101, CIFAR-10, CIFAR-100, SUN397, Stanford Cars, FGVC Aircraft, Describable Textures, Oxford-IIIT Pets, Caltech-101, Oxford Flowers 102. Tab. 6 is the detailed information of these datasets.

Table 6:  Details of DeCLIP downstream datasets.

| DATASET | CLASSES | TRAIN SIZE | TEST SIZE | EVALUATION METRIC |
|---|---|---|---|---|
| CIFAR10 | 10 | 50,000 | 10,000 | ACCURACY |
| CIFAR100 | 100 | 50,000 | 10,000 | ACCURACY |
| FOOD-101 | 101 | 75,750 | 25,250 | ACCURACY |
| OXFORDIIIT-PETS | 37 | 3,680 | 3,669 | MEAN PER CLASS |
| OXFORD 102 FLOWERS | 102 | 2,040 | 6,149 | MEAN PER CLASS |
| SUN | 397 | 19,850 | 19,850 | ACCURACY |
| STANFORD CARS | 196 | 8,144 | 8,041 | ACCURACY |
| DTD | 47 | 3,760 | 1,880 | ACCURACY |
| CALTECH-101 | 102 | 3,060 | 6,085 | MEAN PER |
| FGVC AIRCRAFT | 100 | 6,667 | 3,333 | MEAN PER CLASS |
| IMAGENET1K | 1000 | 1,281,167 | 50,000 | ACCURACY |

**Implementation details**  We follow CLIP (Radford et al., 2021) to train a logistic regression classifier using L-BFGS, with maximum 1,000 iterations, and report the corresponding metric for each dataset. We determine the L2 regularization strength $\lambda$ using a hyperparameter sweep on the validation sets over the range between $10^{-6}$ and $10^{6}$, with 96 logarithmically spaced steps. To save compute required for the sweeps, we perform a parametric binary search that starts with $\lambda = [10^{-6}, 10^{-4}, 10^{-2}, 1, 10^{2}, 10^{4}, 10^{6}]$ and iteratively halves the interval around the peak until it reaches a resolution of 8 steps per decade. The hyperparameter sweeps are performed on a validation split of each dataset. For the datasets that contains a validation split in addition to from the test split, we use the provided validation set to perform the hyperparameter search, and for the datasets that do not provide a validation split or have not published labels for the test data, we split the training dataset to perform the hyperparameter search and report the performance on the validation data.

## E  PROMPT ENGINEERING

Due to the reason that it's relatively rare in the dataset for the text to be a single word, we use prompts such as `"a photo of a {label}"` for zero-shot classification. For a fair comparison, we use the same prompts as proposed in Radford et al. (2021) for the ImageNet dataset. As shown in Fig 11, the prompts reduce the domain gap between the training dataset and testset, and fully consider the different situations for the picture.

a bad photo of a {label}.
a photo of many {label}.
a sculpture of a {label}.
a photo of the hard to see {label}.
a low resolution photo of the {label}.
a rendering of a {label}.
graffiti of a {label}.
a bad photo of the {label}.
a cropped photo of the {label}.
a tattoo of a {label}.
the embroidered {label}.
a photo of a hard to see {label}.
a bright photo of a {label}.
a photo of a clean {label}.
a photo of a dirty {label}.
a dark photo of the {label}.
a drawing of a {label}.
a photo of my {label}.
the plastic {label}.
a photo of the cool {label}.

a close-up photo of a {label}.
a black and white photo of the {label}.
a painting of the {label}.
a painting of a {label}.
a pixelated photo of the {label}.
a sculpture of the {label}.
a bright photo of the {label}.
a cropped photo of a {label}.
a plastic {label}.
a photo of the dirty {label}.
a jpeg corrupted photo of a {label}.
a blurry photo of the {label}.
a photo of the {label}.
a good photo of the {label}.
a rendering of the {label}.
a {label} in a video game.
a photo of one {label}.
a doodle of a {label}.
a close-up photo of the {label}.
a photo of a {label}.

the origami {label}.
the {label} in a video game.
a sketch of a {label}.
a doodle of the {label}.
a origami {label}.
a low resolution photo of a {label}.
the toy {label}.
a rendition of the {label}.
a photo of the clean {label}.
a photo of a large {label}.
a rendition of a {label}.
a photo of a nice {label}.
a photo of a weird {label}.
a blurry photo of a {label}.
a cartoon {label}.
art of a {label}.
a sketch of the {label}.
a embroidered {label}.
a pixelated photo of a {label}.
itap of the {label}.

a jpeg corrupted photo of the {label}.
a good photo of a {label}.
a plushie {label}.
a photo of the nice {label}.
a photo of the small {label}.
a photo of the weird {label}.
the cartoon {label}.
art of the {label}.
a drawing of the {label}.
a photo of the large {label}.
a black and white photo of a {label}.
the plushie {label}.
a dark photo of a {label}.
itap of a {label}.
graffiti of the {label}.
a toy {label}.
itap of my {label}.
a photo of a cool {label}.
a photo of a small {label}.
a tattoo of the {label}.

Figure 11: The prompts for zero-shot testing.

## F  ADDITIONAL STUDY

Table 7: DeCLIP zero-shot performance of ImageNet top1 on different training datasets.

| BACKBONE | DATASET | DATA SIZE | BATCH SIZE | ZERO-SHOT |
|---|---|---|---|---|
| RESNET50 | CONCEPTUALCAPTIONS | 3M | 2,048 | 27.8 |
| RESNET50 | CONCEPTUAL12M | 11M | 4,096 | 41.0 |
| RESNET50 | YFCC | 15M | 4,096 | 41.9 |
| RESNET50 | DECLIP OPEN-SOURCE DATA | 29M | 6,144 | 49.3 |

**Different Pre-training Datasets**  Data is critical for language-image pre-training task. As shown in Tab. 7, we evaluate our DeCLIP on different sources of datasets. Combining Tab.7 and Fig.1, we can see that when the amount of training data continues to scale up, the zero-shot recognition ability continues to improve as well. In addition, we can see that the open source data has high quality. The 29M open source data can achieve 49.3% zero-shot top1 accuracy on ImageNet through the DeCLIP training paradigm. Our open-source data is an affordable benchmark, which would be beneficial for explorations.

**Our YFCC re-implementation**  Although we use the same number of image-text pairs as the CLIP YFCC-15M, our YFCC data is different from CLIP. We also reproduce the naive CLIP on our YFCC-15M data, which results in 35.9% zero-shot top1 accuracy on ImageNet-1K (see Fig. 8). It is relatively higher than the number in CLIP paper (31.1%). We conjecture the improvements might be caused by the different data cleaning strategies. However, our DeCLIP can achieve 41.9% zero-shot accuracy which is also 6.0% higher than our CLIP re-implementation.

Table 8: DeCLIP zero-shot performance of ImageNet top1 on YFCC datasets. Although we use the same amount of data, our YFCC is different with CLIP YFCC due to the different data cleaning strategies.

| MODEL | BACKBONE | DATASET | DATA SIZE | BATCH SIZE | ZERO-SHOT |
|---|---|---|---|---|---|
| CLIP | RESNET50 | YFCC | 15M | — | 31.3 |
| CLIP (OUR REIMP.) | RESNET50 | YFCC* | 15M | 4,096 | 35.9 |
| **DECLIP** | **RESNET50** | **YFCC*** | **15M** | **4,096** | **41.9(↑+6.0)** |

**Memory usage**  Because of the additional views, our DeCLIP is more memory-consuming. Thanks to the ICLR anonymous review comments: a fairer comparison might be doubling the batch size of CLIP. Flowing the ablation study in Fig. 7, we double the batch size and train CLIP-ResNet50 for 64 epochs. The final result is 22.3% which is still 4.9% lower than our DeCLIP model. We summarize the memory usage, training cost, and the final accuracy as below. All experiments are conducted on CC-3M with 16 V100 GPUs.

Table 9: Ablation on Memory usage.

| MODEL | BATCH SIZE PER GPU | MEMORY (GB) | EPOCHS | COST (GPU HOURS) | ZERO-SHOT |
|---|---|---|---|---|---|
| CLIP-RESNET50 | 128 | 15.8 | 64 | 416 | 21.7 |
| CLIP-RESNET50 | 256 | 24.0 | 64 | 399 | 22.3 |
| DECLIP-RESNET50 | 128 | 22.7 | 32 | 304 | 27.2 |

