# OpenReview forum: "Supervision Exists Everywhere: A Data Efficient Contrastive Language-Image  Pre-training Paradigm"
_ICLR.cc/2022/Conference — ICLR 2022 Poster_

### Official Review · Reviewer_nUjZ · 2021-10-31

**Correctness:** 4
**Technical Novelty And Significance:** 2
**Empirical Novelty And Significance:** 3
**Recommendation:** 6
**Confidence:** 5

**Main Review:**

## Strengths

- (+) The proposed DeCLIP shows better ImageNet zero-shot performances than the original CLIP models with the same parameter size but smaller training data size.
- (+) The ablation study (Table 4) supports that the proposed three objectives are important to the final performances. Also, Figure 7 shows that the convergence speed of DeCLIP is faster than the original CLIP loss.

## Weaknesses

### CLIP numbers and DeCLIP numbers would not be comparable; the CLIP training environment is different from the DeCLIP training environment

I fully understand that CLIP results are almost impossible to reproduce with ordinary infrastructures. It is too expensive to reproduce the original ResNet50x64 results (18 days with 592 V100 GPUs). It takes more than one week for ResNet and ViT DeCLIP training even with 80 V100 GPUs (Appendix C), and the largest RegNet DeCLIP takes three weeks with 160 V100 GPUs. Hence, I fully understand that it is very difficult to report many numbers.

However, since this paper argues that "CLIP is data-hungry and DeCLIP can learn strong representation as much as CLIP with much smaller dataset size", I think that the CLIP results should be the reproduced numbers.

- The original CLIP paper did not investigate the impact of dataset size (except ablation study on the YFCC-15M dataset), and there is no justification of the dataset size of 400M. Hence, I think that arguing "CLIP needs 400M data points" needs to be justified by the following studies (namely, in this submission). In other words, I would like to see "reproduced CLIP results" in Figure 1 using the authors' implementation. Since DeCLIP needs more resources than CLIP (because of additional data augmentations, computation costs, memory size, ...), I think the authors can report CLIP results for 15M / 29M / 56M / 88M as DeCLIP.
- I presume that the reproduced CLIP environment is different from the original CLIP implementation (which is not publicly accessible yet). Table 8 amplifies my assumption; the reproduced CLIP performance is 4.6% better than the original CLIP (31.3 => 35.9). In my opinion, it could be very critical if the performance gains are originated from the DeCLIP implementation itself, not from the proposed objective functions. Hence, as I suggested in the previous bullet point, I would like to see the reproduced CLIP numbers for various dataset sizes.
- Third, we do not have enough knowledge on **88M DeCLIP full data** collected by the authors. I wonder whether the performance improvements are from the data collection, not the proposed method. For example, 88M full data quality could be better than the original 400M dataset. As described in Section C, additional filtering processes, such as Chinese caption filtering, would improve the dataset quality compared to CLIP. It is well-known that learning with fewer clean data often outperforms learning with a large noisy dataset. To avoid such concerns, I think Figure 1, Table 2, and 3 should need additional rows "CLIP training + 88M training data" for each model. In my opinion, because the training datasets are different, CLIP 400M models and DeCLIP 88M models are not directly comparable in zero-shot and linear probe benchmarks.
- Finally, CLIP did not use any data augmentation on the language domain, while DeCLIP uses the EDA augmentation for the multi-view supervision. It is not clear whether the performance improvement by MVS in Table 4 is from the EDA augmentation or the multi-view supervision. Because DeCLIP uses more data points than CLIP (Algorithm 1: Line 10), I wonder if the performance gain is from the data augmentation. CLIP + EDA augmentation experiments may answer my question.

Similarly, I would like to suggest reporting RegNetY-64GF CLIP results in Table 2.

### Additional analyses may need quantitative comparisons

This weakness is not very critical, but the submission will be stronger if the authors can address this issue.

Figure 8 and Figure 9 show the CAM visualizations and nearest neighbor search results by CLIP and DeCLIP. However, these results are very easy to cherry-pick; hence, these results should be supported by the quantitative comparisons.

For CAM comparisons, I would recommend weakly-supervised object localization (WSOL) benchmarks. WSOL does not require any additional training but only CAM score maps. I recommend using Choe et al. https://github.com/clovaai/wsolevaluation for comparing CLIP, and DeCLIP CAM results more concretely.

Nearest neighbor search performances can be easily measured by cross-modal retrieval benchmarks (in a zero-shot manner). I would recommend famous COCO or Flickr caption retrieval benchmarks. If the authors need more dense benchmarks, please refer Parekh et al. https://github.com/google-research-datasets/Crisscrossed-Captions.

Since these benchmarks are available without additional expensive training, I strongly recommend reporting the results using the suggested benchmarks.

## Questions

- Why are there no zero-shot results for other benchmarks? I only can see the linear probe verifications (Figure 2, Table 3).
- In Table 3, DeCLIP ViT-B/32 shows worse linear probe results on ImageNet than CLIP VIT-B/32, despite DeCLIP showing better zero-shot ImageNet results in Table 2. What is the guess by the authors for the results?
- It seems that the proposed DeCLIP needs more computation / memory resources compared to CLIP. Could the authors provide the memory consumption comparison between CLIP and DeCLIP? I presume that with the same GPUs, DeCLIP cannot use the same batch size as CLIP.

## List of suggested experiments

I fully understand that the rebuttal period is too short to produce the following numbers. At least, I would expect 88M CLIP results (Table 2) for ResNet-50 and ViT-B/32.

- I am willing to revise my score if these results show the same tendency as the paper claim
  - Figure 1. Reproduced CLIP ImageNet zero-shot top-1 results for 15M / 29M / 56M / 88M (15M is already in Table 8) -- ResNet-50
  - Table 2. Reproduced CLIP ImageNet zero-shot top-1 results for 88M with ResNet-50, (ResNet-101 -- not necessary), ViT-B/32, and RegNETY-64GF
  - Table 3. Reproduced CLIP linear probe results on 11 downstream tasks
- These numbers are not mandatory, but I think these results will make the submission stronger
  - Figure 8. WSOL benchmark (https://github.com/clovaai/wsolevaluation)
  - Figure 9. Cross-modal retrieval benchmarks (COCO Caption, Flickr Caption, Crisscrossed Caption, ...)
  - CLIP + EDA augmentation results on 15M / 29M / 56M / 88M datasets (ImageNet zero-shot top-1)

**Summary Of The Paper:**

This paper proposes a data-efficient CLIP (DeCLIP) by employing three additional objectives:

- (1) self-supervision within each modality (e.g., vision domain => SimSiam, text domain => masked language model as BERT)
- (2) Multi-view supervision (e.g., resize random crop for vision modality and EDA augmentation -- synonym replacement, random insertion, random swap, and
random deletion -- for text modality)
- (3) Nearest-neighbor supervision (e.g., using close plausible neighbors to pseudo-positives)

Table 2 shows that DeCLIP shows better zero-shot ImageNet top-1 accuracies with smaller dataset sizes (400M => 88M), and Table 3 shows that DeCLIP shows better overall linear probe performances on various datasets.

Also, Table 4 and Figure 7 support that the proposed three additional objectives make the CLIP training much efficient than the vanilla version.

**Summary Of The Review:**

This paper shows a strong empirical contribution; DeCLIP shows better results than CLIP with a smaller dataset size. However, I think the weakness of this paper slightly overweighs the strengths.

In particular, I am not fully convinced that DeCLIP is data-efficient and CLIP is data-inefficient. This argument should be supported by empirical comparisons, e.g., the reproduced CLIP results should be reported in Figure 1, Table 2, and 3. I listed why I think the DeCLIP results and the CLIP results are not directly comparable in the weakness section.

All technical components of DeCLIP are hard to say novel; self-supervision uses SimSiam and masked language model, multi-view supervision uses the EDA augmentation, nearest neighbor supervision can be viewed as a self-distillation method. However, the combination of known techniques can be novel if the empirical contribution is significant. I think this paper has a limited borderline novelty, but the empirical contribution can be a strength. For example, if this paper outperforms "ViT-L/14@336px" with much fewer data points, I think the empirical contribution exceeds its weakness. (disclaimer: I do not request ViT-L/14@336px results. I know that it is very expensive and unachievable by ordinary infrastructures) As of now, I think the empirical contribution may need more verifications, especially for the vanilla CLIP results.

Overall, I recommend "marginally below the acceptance threshold" for my initial recommendation.

-----------------------------
Post rebuttal comment.

After reading the responses (for raised concerns by all reviewers) and the revised paper, I think the authors address my concerns very well in general. I would like to encourage the authors includes the results in A3, A4 (not completed yet), A5, A6 in the paper. Also, I encourage the authors to add zero-shot results (Q7) in the final revision, which was not possible at the submission time (I missed it).

After the revision, the arguments are generally well-supported (revised my score for correctness has been updated to 4 from 3). Now, I recommend "6: marginally above the acceptance threshold" for this paper.

---

> ### Author Response · Authors · 2021-11-22
> **Responses to Reviewer nUjZ (Part I)**
>
> We sincerely thank you for the detailed feedback. We will explain your concerns point by point.
>
> >**Q1: ... I would like to see "reproduced CLIP results" in Figure 1 using the authors' implementation. Since DeCLIP needs more resources than CLIP (because of additional data augmentations, computation costs, memory size, ...), I think the authors can report CLIP results for 15M / 29M / 56M / 88M as DeCLIP.**
>
> A1: We train the CLIP-ResNet50 baseline on 29M / 56M / 88M in the rebuttal. As shown in the following table, our DeCLIP consistently outperforms the CLIP across all dataset sizes. We also train the CLIP-ViT-B/32 on the 88M dataset.
> Interestingly, CLIP-ViT-B/32 benefits more from our proposed methods (+8.8\% compared with +5.6\% for CLIP-ResNet50). We conjecture that this is because ViT is more data-hungry than CNN.
> We have updated Fig.1, Tab.2 in our revision as well.
>
> | Dataset         | 3M         | 15M        | 29M        | 56M        | 88M        |
> |-----------------|------------|------------|------------|------------|------------|
> | CLIP-ResNet50   | 20.6       | 35.9       | 44.2       | 54.5       | 56.9       |
> | DeCLIP-ResNet50 | 27.2(+6.6) | 41.9(+6.0) | 49.3(+5.1) | 60.4(+5.9) | 62.5(+5.6) |
> | CLIP-ViT-B/32    |            |            |            |            | 57.4       |
> | DeCLIP-ViT-B/32  |            |            |            |            | 66.2(+8.8) |
>
> >**Q2: ... I think Figure 1, Table 2, and 3 should need additional rows "CLIP training + 88M training data" for each model. In my opinion, because the training datasets are different, CLIP 400M models and DeCLIP 88M models are not directly comparable in zero-shot and linear probe benchmarks. ...**
>
> A2:  We have updated Fig.1, Tab.2/3 in the revision. In terms of linear probe benchmarks, we transfer our reproduced 'CLIP-88M' models into 11 downstream datasets using the same protocol. As shown in the following table, our DeCLIP models outperform CLIP counterparts by a large margin, a 3.4\%/4.6\% average improvement for DeCLIP-ResNet50 and DeCLIP-ViT-B/32 respectively.
>
> | Model               | Pets | C10  | C100 | SUN  | F101 | Flow | Cars | Cal. | Air. | DTD  | IN   | AVG. |
> |---------------------|------|------|------|------|------|------|------|------|------|------|------|------|
> | CLIP-88M-ResNet50   | 85.1 | 87.3 | 65.0 | 71.3 | 80.8 | 98.2 | 77.2 | 89.6 | 44.8 | 71.0 | 70.8 | 76.5 |
> | DeCLIP-88M-ResNet50 | 88.7 | 89.8 | 71.2 | 72.8 | 82.7 | 99.2 | 81.7 | 93.9 | 48.4 | 76.8 | 74.0 | 79.9(+3.4) |
> | CLIP-88M-ViT-B/32    | 85.3 | 91.6 | 74.2 | 72.3 | 80.8 | 97.9 | 77.6 | 93.2 | 47.0 | 72.5 | 70.3 | 78.4 |
> | DeCLIP-88M-ViT-B/32  | 89.2 | 96.5 | 84.7 | 75.0 | 85.0 | 99.1 | 81.6 | 94.8 | 53.5 | 78.5 | 75.3 | 83.0(+4.6) |
>
> >**Q3: ... It is not clear whether the performance improvement by MVS in Table 4 is from the EDA augmentation or the multi-view supervision. Because DeCLIP uses more data points than CLIP (Algorithm 1: Line 10), I wonder if the performance gain is from the data augmentation. CLIP + EDA augmentation experiments may answer my question. ...**
>
> A3: EDA (text augmentation) is a crucial part of multi-view supervision(MVS), because we need augmentation to provide richer supervision. We do 'CLIP + EDA' ablation study in rebuttal to further disentangle the effects of EDA and MVS. As we can see, EDA is beneficial for CLIP, but the gain is relatively minor than the entire MVS. Due to time and resource constraints, we only do the experiments on the CC-3M dataset, but we believe the trend would be the same for different data sizes.
>
> |                     | CLIP | CLIP + text EDA | CLIP +MVS | DeCLIP |
> |---------------------|------|-----------------|-----------|--------|
> | Zero-shot acc. @ 3M | 20.6 | 22.1            | 24.8      | 27.2   |
>
>
> >**Q4: Similarly, I would like to suggest reporting RegNetY-64GF CLIP results in Table 2.**
>
> \textbf{A4: } We only performed one training trial for the 'RegNETY-64GF' size model due to the training cost. Unfortunately, we cannot reproduce the RegNetY-64GF CLIP results during rebuttal due to time and resource constraints. We hope our ablation study in Q1 will ease your concern about our proposed objective functions.
>
>
> >**Q5:  For CAM comparisons, I would recommend weakly-supervised object localization (WSOL) benchmarks. WSOL does not require any additional training but only CAM score maps.**
>
> A5:  Thanks for pointing out this valuable test tool. Following WSOL(https://github.com/clovaai/wsolevaluation), we test CAM score maps of DeCLIP-ResNet50, our reproduced CLIP-ResNet50, and ImageNet-ResNet50 on the ImageNet-V2 benchmark provided by Choe et al. As we can see from the following table, our DeCLIP has a better performance compared to counterparts.
>
> |                         | CLIP-ResNet50 | DeCLIP-ResNet50 | Imagenet-ResNet50 |
> |-------------------------|---------------|-----------------|-------------------|
> | MaxBoxAcc. @ ImagenetV2 | 53.8          | 56.5            | 54.5              |

---

> > ### Author Response · Authors · 2021-11-22
> > **Responses to Reviewer nUjZ (Part II)**
> >
> > >**Q6:  Nearest neighbor search performances can be easily measured by cross-modal retrieval benchmarks (in a zero-shot manner). I would recommend famous COCO or Flickr caption retrieval benchmarks.**
> >
> > A6: The CLIP paper does not report the retrieval performance of CLIP-ResNet50 nor CLIP-ViT-B/32. In the rebuttal, we test the image text retrieval on popular benchmark MSCOCO. We reuse the 'CLIP-88M' models in A1. As shown in the following table, our DeCLIP models outperform CLIP counterparts on text retrieval and image retrieval tasks. Worth mentioning, our results are lower than that of CLIP-ViT-L/14@336px due to model capacity and data amount. We are now training a larger ViT model on a larger dataset, the results will be updated in the near future.
> >
> > | MSCOCO                   |      | I2T  |      |      | T2I  |      |
> > |--------------------------|------|------|------|------|------|------|
> > |                          | R1   | R5   | R10  | R1   | R5   | R10  |
> > | CLIP-400M-ViT-L/14@336px | 58.4 | 81.5 | 88.1 | 37.8 | 62.4 | 72.2 |
> > | CLIP-88M-ResNet50        | 30.3 | 56.2 | 67.4 | 19.1 | 40.8 | 52.4 |
> > | CLIP-88M-ViT-B/32        | 31.8 | 57.7 | 69.2 | 20.0 | 42.9 | 55.2 |
> > | DeCLIP-88M-ResNet50      | 31.1 | 57.4 | 68.8 | 20.0 | 43.0 | 55.0 |
> > | DeCLIP-88M-ViT-B/32      | 33.1 | 59.4 | 71.5 | 22.1 | 45.7 | 58.1 |
> >
> >
> > >**Q7:  Why are there no zero-shot results for other benchmarks? I only can see the linear probe verifications (Figure 2, Table 3).**
> >
> > A7:  CLIP did not release the class names and prompts of downstream tasks until Sep. 23, which was only ten days before the ICLR submission deadline. Due to time constraints and sophisticated zero-shot evaluation protocols for 11 datasets, we are sorry we cannot provide these results at this time.
> >
> >
> > >**Q8:  In Table 3, DeCLIP ViT-B/32 shows worse linear probe results on ImageNet than CLIP VIT-B/32, despite DeCLIP showing better zero-shot ImageNet results in Table 2. What is the guess by the authors for the results?**
> >
> > A8:  This is an interesting finding. We conjuncture that DeCLIP has a powerful text encoder because our proposed supervision also benefits the text encoder. When it comes to linear probe, we will drop the text encoder, which hurts the performance of DeCLIP.
> >
> >
> > >**Q9:  Could the authors provide the memory consumption comparison between CLIP and DeCLIP? I presume that with the same GPUs, DeCLIP cannot use the same batch size as CLIP.**
> >
> > A9:  Flowing the ablation study in Fig.7, we double the batch size and train CLIP-ResNet50 for 64 epochs. The final result is 22.3\% which is still 4.9\% lower than our DeCLIP model. We summarize the memory usage, training cost, and the final accuracy as below. All experiments are conducted on CC-3M with 16 V100 GPUs.
> >
> > | Model           | batch size per GPU | memory usage (GB) | Epochs | training cost (GPU hours) | zero-shot top1 |
> > |-----------------|--------------------|-------------------|--------|---------------------------|----------------|
> > | CLIP-ResNet50   | 128                | 15.8              | 64     | 416                       | 21.7           |
> > | CLIP-ResNet50   | 256                | 24.0              | 64     | 399                       | 22.3           |
> > | DeCLIP-ResNet50 | 128                | 22.7              | 32     | 304                       | 27.2           |
> >
> >
> > >**Q10:  ... For example, if this paper outperforms "ViT-L/14@336px" with much fewer data points, I think the empirical contribution exceeds its weakness. ...**
> >
> > A10:  This is an excellent point. We only performed one training trial for the 'RegNETY-64GF' size model due to the training cost. In the future, we hope to train a powerful Vision Transformer (e.g. ViT-L/14) with more data (e.g. 400M), which we expect to have a SOTA result.

---

> > > ### Comment · Reviewer_nUjZ · 2021-11-26
> > > **Thanks for the responses**
> > >
> > > Dear Authors,
> > >
> > > Thanks for addressing my concerns very well. I checked the revision as well. It seems that my concerns are mostly covered by the responses and the revised paper.
> > >
> > > I would like to encourage the authors includes the results in A3, A4 (not completed yet), A5, A6 in the paper. Also, I encourage the authors to add zero-shot results (Q7) in the final revision. Thanks for the clarification in A7, I totally forgot about this. I think the response makes sense and is reasonable. I also slightly encourage adding discussion related to Q8 if it is possible.
> > >
> > > I would like to revise my score to "6: marginally above the acceptance threshold". Thanks for the heavy additional experiments.

---

### Official Review · Reviewer_x3bM · 2021-11-02

**Correctness:** 3
**Technical Novelty And Significance:** 3
**Empirical Novelty And Significance:** 3
**Recommendation:** 6
**Confidence:** 3

**Main Review:**


Pro:
This is the first work that I can think of that successfully combines self-supervision and supervised training, at such a large scale. While self-supervised training has shown promising results in CV, the experiments are usually done in an unsupervised/self-supervised setting where the model is trained with only raw images and then tested with linear probes. It remains less explored how one could combing self-supervision and supervised image-text pairs data and whether self-supervision is still helpful when a large amount of supervised data are available. This work provides strong empirical evidence that self-supervision can be combined with supervised data.



Con:
Table 2 shows that DeCLIP with 88M data can outperform CLIP with 400M. However, the 88M data are collected by the authors while the 400M data are collected by CLIP authors. As shown in Table 8 in the appendix, how one process/selects the data can result in a large performance difference.

I cannot safely conclude that the data efficiency/performance improvement shown in Table 2 is completely attributed to DeCLIP. It could be attributed partially to that the 88M data are a high-quality subset of the 400M data. I would recommend adding a baseline of CLIP trained with 88M data.

The authors provide fair baselines of CLIP in Figure 7 and Table 8 (in the appendix) but the dataset scale is limited. As one of the core claims is the benefit of DeCLIP under large-scale pre-training, I think providing fair baselines in Table 2 is important.

Minor points:

   1. Analysis lacking on why we need self-supervision signals / nearest-neighbor supervision.

      As noted in Pro, this paper touches upon an important problem: the relationship between self-supervision and supervised data.

      However, the authors only show empirical evidence of self-supervision can be helpful even when a large amount of supervised data are available. It would be helpful to provide some discussions. For example, does cross-view self-supervision provide signals simply not available in supervised data? Does DeCLIP do better in certain types of tasks?

   2. I am under the impression that nearest neighbors are usually used to provide "hard negatives" in contrastive learning (e.g., in the Image Retrieval task in UNITER (Chen et al., 2019)). However, here the nearest neighbors are used as "positive" examples. I wonder what causes the different practices.



**Summary Of The Paper:**

The paper proposes DeCLIP to further utilize the data potential by adding three training objectives to CLIP pre-training: 1) inspired by SimSiam and BERT, self-supervised objectives are added to both image and text; 2) they generate different views for both images and text, and apply contrastive objectives; 3) they sample neighbor text as additional positive examples.

DeCLIP improves data efficiency. With web-crawled data, DeCLIP outperforms CLIP counterparts with 4.5x smaller amount of data. In addition, while introducing addition objectives and especially different views increases per-batch compute time by 1.5x, the authors show that DeCLIP still outperforms CLIP when given the same compute time budget.



**Summary Of The Review:**

**After rebuttal: I have read the authors' rebuttal and it resolves my concerns. I will raise my score to between 6 and 8.**

---

> ### Author Response · Authors · 2021-11-22
> **Responses to Reviewer x3bM**
>
> We sincerely thank you for your comprehensive comments and constructive advices. We will explain your concerns point by point.
>
> >**Q1:  ... I would recommend adding a baseline of CLIP trained with 88M data. The authors provide fair baselines of CLIP in Figure 7 and Table 8 (in the appendix) but the dataset scale is limited. As one of the core claims is the benefit of DeCLIP under large-scale pre-training, I think providing fair baselines in Table 2 is important.**
>
> A1:  We train the CLIP-ResNet50 baseline on 29M / 56M / 88M in the rebuttal. As shown in the following table, our DeCLIP consistently outperforms the CLIP across all dataset sizes. We also train the CLIP-ViT-B/32 on the 88M dataset.
> Interestingly, CLIP-ViT-B/32 benefits more from our proposed methods (+8.8\% compared with +5.6\% for CLIP-ResNet50). We conjecture that this is because ViT is more data-hungry than CNN.
> We have updated Fig.1, Tab.2 in our revision as well.
>
>
> | Dataset         | 3M         | 15M        | 29M        | 56M        | 88M        |
> |-----------------|------------|------------|------------|------------|------------|
> | CLIP-ResNet50   | 20.6       | 35.9       | 44.2       | 54.5       | 56.9       |
> | DeCLIP-ResNet50 | 27.2(+6.6) | 41.9(+6.0) | 49.3(+5.1) | 60.4(+5.9) | 62.5(+5.6) |
> | CLIP-ViT-B/32    |            |            |            |            | 57.4       |
> | DeCLIP-ViT-B/32  |            |            |            |            | 66.2(+8.8) |
>
> >**Q2:  ... the authors only show empirical evidence of self-supervision can be helpful even when a large amount of supervised data are available. It would be helpful to provide some discussions. For example, does cross-view self-supervision provide signals simply not available in supervised data? Does DeCLIP do better in certain types of tasks?**
>
> A2:  Understanding how self-supervision benefits supervised learning is an interesting topic. As detailed in SupCon[1], contrastive self-supervision can bring more positives to supervised cross-entropy loss, which is helpful for classification. Our DeCLIP shares the same spirit as SupCon: SS, MVS, and NNS could all be recognized as providing more positive image-text pairs. Because of more positive pairs, we could learn more invariant alignment between image and text, leading to better performance.
>
>
> >**Q3:  I am under the impression that nearest neighbors are usually used to provide "hard negatives" in contrastive learning (e.g., in the Image Retrieval task in UNITER (Chen et al., 2019)). However, here the nearest neighbors are used as "positive" examples. I wonder what causes the different practices.**
>
> A3:  UNITER (Chen et al., 2019) uses nearest neighbors (NN) as hard negatives in fine-tuning stage (for image-text retrieval task), while our DeCLIP uses NN as positive examples in the pre-training stage.
> Intuitively, fine-tuning aims to fit the given positive/negative pairs more precisely; hence 'hard negatives' in fine-tuning are true negatives that could help push the hard pairs away.
> However, in our DeCLIP pre-training phase, we aim to achieve a more general representation via a weakly supervised objective. One image might likely have several appropriate text captions (as shown in Fig.3 and Fig.9). Our NN supervision can be regarded as a semantic-level augmentation to improve the generalization.
>
> >**References:**
>
>
> [1] Khosla, Prannay, et al. "Supervised contrastive learning." arXiv preprint arXiv:2004.11362 (2020).

---

### Official Review · Reviewer_kziT · 2021-11-03

**Correctness:** 4
**Technical Novelty And Significance:** 3
**Empirical Novelty And Significance:** 3
**Recommendation:** 8
**Confidence:** 3

**Main Review:**

Strength:
- Extensive experiments show DeCLIP achieves competitive improvements over CLIP while uses much less pre-training data, which also results in much less training time. With the same amount of pre-training data, DeCLIP achieves significant improvements over CLIP (+7%)
- The three additions are simple yet effective. The ablation study shows that all three of them contribute to the final performance improvements.
- Visualization show that nearest neighbor pairs can provide reasonable similar supervision signal compared to the original pair.
- The paper is well-written and easy to follow.


A few questions about the paper:
- For nearest neighbor pairs, are them all sampled from the same dataset as the original image-text pair? If so, are there different FIFO queue Q for different datasets during pre-training?
- What about other distance measures to compute nearest neighbors, for example cosine similarity?
- Why not also sample nearest neighbor images? If added, it should add 8x more supervision than the original CLIP.


**Summary Of The Paper:**

This paper proposes DeCLIP with three additions to the original CLIP model: (1) single-modal self supervised pre-training; (2) cross-modal contrastive pretraining across multiple views and (3)  cross-modal contrastive pretraining across nearest neighbors in feature space. DeCLIP is able to achieve higher performance on most of downstream datasets with much less pre-training data (80M vs. 400M).

**Summary Of The Review:**

This paper provides three simple yet effective additions to CLIP and show its strong performance with much less pre-training data. Overall, I feel the paper is clearly written, with extensive experimental results to support its claim.

---

> ### Author Response · Authors · 2021-11-22
> **Responses to Reviewer kziT**
>
> Thank you for your appreciation of our paper. We are glad to answer your questions point by point.
>
> >**Q1:  For nearest neighbor pairs, are them all sampled from the same dataset as the original image-text pair?**
>
> A1:  No, the nearest neighbor pairs are from the entire training dataset. For example, a pair from the CC dataset might get its nearest neighbor from the YFCC dataset, which provides richer supervisory signals.
>
> >**Q2:  If so, are there different FIFO queue Q for different datasets during pre-training?**
>
> A2:  We maintain only one FIFO queue Q for the entire training dataset. Considering that the goal of the queue is to simulate the entire dataset distribution, it is favorable to maintain only one queue.
>
> >**Q3: What about other distance measures to compute nearest neighbors, for example cosine similarity?**
>
> A3:  We encountered an embarrassing condition. After double-checking our implementation, we find we use cosine similarity as distance measures instead of Euclidean distance. This is because the NNCLR [1] paper claims Euclidean distance, but its open-source implementation [2] adopts cosine similarity. Thanks for pointing it out, and we have corrected it in the revision. We also try the Euclidean distance in the rebuttal. These two distance metrics do not have significant differences regarding the final accuracy.
>
>
> >**Q4:  Why not also sample nearest neighbor images? If added, it should add 8x more supervision than the original CLIP.**
>
> A4:  This choice is result-driven: image NN does not bring improvement in our experiments. We try to answer it by visualizing the image NN pairs. It turns out that image NN is likely to provide images with a similar color or the same category to the main object, which might not be suitable for the original text.
>
> >**References:**
>
> [1] Dwibedi, Debidatta, et al. "With a little help from my friends: Nearest-neighbor contrastive learning of visual representations." arXiv preprint arXiv:2104.14548 (2021).
>
> [2] https://github.com/facebookresearch/vissl/issues/316

---

### Official Review · Reviewer_a3Wz · 2021-11-03

**Correctness:** 4
**Technical Novelty And Significance:** 3
**Empirical Novelty And Significance:** 2
**Recommendation:** 6
**Confidence:** 4

**Main Review:**

Pros:
- Clear advantage of the proposed framework on data efficiency.
- Simple and straightforward methodology.
- Carefully designed experiments. (Comparison to CLIP with 15M data and ablation on computation time.)

Cons:
- Limited novelty in terms of the method:
    - The self-supervision and multi-view supervision are very similar to what has been used in Yuan et al.
    - The Nearest-neighbor supervision was used in Dwibedi et al. and Van Gansbeke et al.
- Experiments:
    - Missing experiments:
        - One of the biggest advantages of CLIP is its robustness to domain shift. Can the authors also provide performance on datasets like ImageNet-R, ImageNet Sketch etc.?
        - No results on image text retrieval.
        - The CLIP reports on 27 downstream datasets. Is there any specific reason for not comparing on all these datasets?
    - While the authors compare with training time, another big difference is in memory consumption. With the additional views, it is equivalent to doubling the batch size. A fairer comparison should be doubling the batch size for CLIP training and train for the same number of iterations or the same amount of time. Since large batch size is important in contrastive learning, it is worth-doing to me.

Questions:
- I believe CLIP also use random crop during training. How do the authors design the small local view (how small)? Can authors provide ablations on the effect of small size?
    - In addition, is the augmented view the same size (in terms of network input) as the original view but just more zoomed in?
- Did the authors do careful duplication removal?

**Summary Of The Paper:**

In this paper, the authors mitigate the data-hungriness of the CLIP model. The authors propose three directions: single-modality self-supervision; multi-view multi-modality contrastive learning, and nearest-neighbor supervision. With the proposed three components, the authors can achieve better or comparable results with CLIP with more than 4x fewer data.

**Summary Of The Review:**

While the novelty is limited, the authors are able to achieve good results with a much smallerset data size. Potentially, with more data, this method can achieve much more superior performance than CLIP. However, I think the insights conveyed from this paper are limited so I will choose borderline accept.


Post-rebuttal update:
- I am very happy with the additional results, which address most of my concerns on fair comparison with CLIP. I will remain the original rating due to the limited "novelty". But I would give a 7 if there is a 7 because I think this paper should be accepted.

---

> ### Author Response · Authors · 2021-11-22
> **Responses to Reviewer a3Wz(Part I)**
>
>
> We sincerely thank you for the valuable comments on our paper. We will explain your concerns point by point.
>
> >**Q1:  The self-supervision and multi-view supervision are very similar to what has been used in Yuan et al.**
>
> A1:  We share the same spirit as Yuan et al. We have two differences:
> 1) For self-supervision, Yuan et al. use contrastive loss (MoCo) for text self-supervision, while we use masked language modeling for text self-supervision, which we find more effective than contrastive loss.
> 2) Yuan et al. are limited to the small clean COCO dataset, while we are the first to use self-supervision and multi-view supervision in the 88-million-scale image-text pre-training task.
>
>
> >**Q2:  The Nearest-neighbor supervision was used in Dwibedi et al. and Van Gansbeke et al.**
>
> A2:  As detailed in Sec 2.2, Dwibedi et al. and Van Gansbeke et al. are limited in a single visual modality. They use Nearest-Neighbors (NN) positives from other images in the ImageNet dataset. We firstly introduce NN into the multi-modal setting. We use text NN from other pairs to serve as the positive pair with the original image (examples in Fig.3).
>
>
>
> >**Q3:  One of the biggest advantages of CLIP is its robustness to domain shift. Can the authors also provide performance on datasets like ImageNet-R, ImageNet Sketch, etc.?**
>
> A3:  The CLIP paper does not report the performance on ImageNet-R, ImageNet-Sketch of CLIP-ResNet50, nor ResNet-ViT-B/32 (it only has the performance of the biggest CLIP-ViT-L/14@336px).
> We train CLIP on our 88M data for a fair comparison, denoted as 'CLIP-88M'. As shown in the following table, our DeCLIP models achieve better performance than CLIP counterparts on domain-shift datasets, outperforming ImageNet-ResNet101. It is also interesting to find that DeCLIP-ViT performs much better on domain-shift datasets than DeCLIP-ResNet.  Worth mentioning, our performance is still lower than that of CLIP-ViT-L/14@336px due to the model capacity and data amount. We believe that with increased model size, potentially with larger ViT, our DeCLIP could outperform CLIP.
>
> |                     | ImageNet | ImageNet-R | ImageNet Sketch | ImageNet V2 |
> |:-------------------:|:--------:|:----------:|:---------------:|:-----------:|
> |      ImageNet-ResNet101      |   76.2   |    37.7    |       25.2      |     64.3    |
> |  CLIP-400M-ViT-L/14@336px |   76.2   |    88.9    |       60.2      |     70.1    |
> |  CLIP-88M-ResNet50  |   56.9   |     42     |       31.5      |     60.2    |
> | DeCLIP-88M-ResNet50 |   62.5   |     54     |       40.8      |     66.2    |
> |   CLIP-88M-ViT-B/32  |   57.4   |    46.9    |       32.9      |     61.7    |
> |  DeCLIP-88M-ViT-B/32 |   66.2   |    64.8    |       48.7      |     70.7    |
>
>
> >**Q4:  No results on image text retrieval.**
>
> A4:  The CLIP paper does not report the retrieval performance of CLIP-ResNet50 nor CLIP-ViT-B/32. In the rebuttal, we test the image text retrieval on popular benchmark MSCOCO. We reuse the 'CLIP-88M' models in A3. As shown in the following table, our DeCLIP models outperform CLIP counterparts on text retrieval and image retrieval tasks. Again, our results are lower than that of CLIP-ViT-L/14@336px due to model capacity and data amount. We are now training a larger ViT model on a larger dataset, the results will be updated in the near future.
>
>
> | MSCOCO                   |      | I2T  |      |      | T2I  |      |
> |--------------------------|------|------|------|------|------|------|
> |                          | R1   | R5   | R10  | R1   | R5   | R10  |
> | CLIP-400M-ViT-L/14@336px | 58.4 | 81.5 | 88.1 | 37.8 | 62.4 | 72.2 |
> | CLIP-88M-ResNet50        | 30.3 | 56.2 | 67.4 | 19.1 | 40.8 | 52.4 |
> | CLIP-88M-ViT-B/32        | 31.8 | 57.7 | 69.2 | 20.0 | 42.9 | 55.2 |
> | DeCLIP-88M-ResNet50      | 31.1 | 57.4 | 68.8 | 20.0 | 43.0 | 55.0 |
> | DeCLIP-88M-ViT-B/32      | 33.1 | 59.4 | 71.5 | 22.1 | 45.7 | 58.1 |

---

> > ### Author Response · Authors · 2021-11-22
> > **Responses to Reviewer a3Wz (Part II)**
> >
> > >**Q5:  The CLIP reports on 27 downstream datasets. Is there any specific reason for not comparing on all these datasets?**
> >
> > A5:  We begin with the 12 datasets from the well-studied evaluation suite introduced by Kornblith et al., 2019 [1]. Other 15 datasets are maintained by the CLIP team, some of them are created by CLIP, such as Country211 and Rendered SST2 (A.1. Datasets in CLIP paper).
> > Within these 12 datasets, Birdsnap can not be downloaded, and PASCAL VOC 2007 classification is replaced by more challenging ImageNet-1K, resulting in our 11 datasets.
> >
> > >**Q6: ... A fairer comparison should be doubling the batch size for CLIP training and train for the same number of iterations or the same amount of time ...**
> >
> > A6:  Flowing the ablation study in Fig.7, we double the batch size and train CLIP-ResNet50 for 64 epochs. The final result is 22.3\% which is still 4.9\% lower than our DeCLIP model. We summarize the memory usage, training cost, and the final accuracy as below. All experiments are conducted on CC-3M with 16 V100 GPUs.
> >
> > | Model           | batch size per GPU | memory usage (GB) | Epochs | training cost (GPU hours) | zero-shot top1 |
> > |-----------------|--------------------|-------------------|--------|---------------------------|----------------|
> > | CLIP-ResNet50   | 128                | 15.8              | 64     | 416                       | 21.7           |
> > | CLIP-ResNet50   | 256                | 24.0              | 64     | 399                       | 22.3           |
> > | DeCLIP-ResNet50 | 128                | 22.7              | 32     | 304                       | 27.2           |
> >
> >
> > >**Q7:  I believe CLIP also use random crop during training. How do the authors design the small local view (how small)? Can authors provide ablations on the effect of small size?**
> >
> > A7:  As shown in issue \#33 of the official CLIP GitHub repo, CLIP first resizes the image to 224 in its minimum dimension and then randomly crops a square 224x224; thus, it is a global view. As detailed in the image augmentations in Appendix B, we use Pytorch builtin RandomResizedCrop with scale in [0.2,1.0]. We can crop as small as 20\% of the original image and resize it into 224*224. To further verify the small size effect, we do the ablation by replacing the RandomResizedCrop to [1.0,1.0] of our DeCLIP. We train the DeCLIP-ResNet50 w/o small size crop on CC-3M for 32 epochs. The final result is 24.1\% on which is 3.1\% lower than our DeCLIP w/ small size crop.
> >
> > >**Q8:  In addition, is the augmented view the same size (in terms of network input) as the original view but just more zoomed in?**
> >
> > A8:  Yes, the input image size of all our DeCLIP models is 224.
> >
> > >**Q9: Did the authors do careful duplication removal?**
> >
> > A9: Thanks for pointing it out. We did not do careful duplication removal in this version. We will do careful duplication in our future version with more data and a larger model.
> >
> > >**References:**
> >
> > [1] Kornblith, S., Shlens, J., and Le, Q. V. Do better imagenet models transfer better? In Proceedings of the IEEE conference on computer vision and pattern recognition, pp. 2661–2671, 2019.

---

### Author Response · Authors · 2021-11-22
**Summary of revision**

Dear reviewers, thanks for the detailed feedback and constructive advice! We clarify the following changes in the revision.


 + Update Fig.1 with our reimplemented CLIP-ResNet50 results on 15M/29M/56M/88M datasets.

 + Update Tab.2/3 with our reimplemented CLIP-ResNet50 and CLIP-ViT-B/32 results on the 88M dataset.

 + Add memory usage ablation study in Appendix F.

 + Add the reason why we use 11 downstream tasks in Appendix D.

  + Fix writing mistakes: NN measure methods should be cosine similarity; the data size of ALIGN should be 1.8B.

We hope we have addressed all of your concerns. Discussions are always open. Thank you!

---

### Decision · Program_Chairs · 2022-01-20

**Decision:**

Accept (Poster)

**Comment:**

This paper aims at improving the data efficiency of pretraining in CLIP. This is a practically meaningful research direction. The proposed method is simple, even kind of straightforward and has limited innovations. It combines self-supervision within each modality, multi-view supervision across modalities, and nearest-neighbor supervision from other similar pairs. Such a combination showed strong empirical results: achieved better performance using seven times fewer data. The rebuttals resolved most critical concerns on experiments, such as fair comparisons with the original CLIP work.